# STING induces early IFN-β in the liver and constrains myeloid cell-mediated dissemination of murine cytomegalovirus

Pia-Katharina Tegtmeyer[1,7], Julia Spanier [1,7], Katharina Borst[1,4], Jennifer Becker[1], André Riedl[2], Christoph Hirche[1], Luca Ghita[1], Jennifer Skerra[1], Kira Baumann[1], Stefan Lienenklaus [1,5], Marius Doering [1,6], Zsolt Ruzsics[2] & Ulrich Kalinke [1,3]

Cytomegalovirus is a DNA-encoded β-herpesvirus that induces STING-dependent type 1 interferon responses in macrophages and uses myeloid cells as a vehicle for dissemination. Here we report that STING knockout mice are as resistant to murine cytomegalovirus (MCMV) infection as wild-type controls, whereas mice with a combined Toll-like receptor/ RIG-I-like receptor/STING signaling deficiency do not mount type 1 interferon responses and succumb to the infection. Although STING alone is dispensable for survival, early IFN-β induction in Kupffer cells is STING-dependent and controls early hepatic virus propagation. Infection experiments with an inducible reporter MCMV show that STING constrains MCMV replication in myeloid cells and limits viral dissemination via these cells. By contrast, restriction of viral dissemination from hepatocytes to other organs is independent of STING. Thus, during MCMV infection STING is involved in early IFN-β induction in Kupffer cells and the restriction of viral dissemination via myeloid cells, whereas it is dispensable for survival.

[1] TWINCORE–Centre for Experimental and Clinical Infection Research, a joint venture between the Hanover Medical School and the Helmholtz Centre for Infection Research, Institute for Experimental Infection Research, 30625 Hanover, Germany. [2] Faculty of Medicine, Medical Center, Institute for Virology, University of Freiburg, 79106 Freiburg, Germany. [3] Cluster of Excellence-Resolving Infection Susceptibility (RESIST), Hanover Medical School, 30625 Hanover, Germany. [4] Present address: Faculty of Medicine, Institute of Neuropathology, University of Freiburg, 79106 Freiburg, Germany. [5] Present address: Central Animal Facility, Imaging-Center, Hanover Medical School, 30625 Hanover, Germany. [6] Present address: Institute Curie, Immunity and Cancer Department, PSL Research University, INSERM U932, 75005 Paris, France. [7] These authors contributed equally: Pia-Katharina Tegtmeyer, Julia Spanier. Correspondence and requests for materials should be addressed to U.K. (email: Ulrich.Kalinke@twincore.de)

Human cytomegalovirus (HCMV) belongs to the subfamily of beta-herpesviruses and is widely spread in the global population. Immunocompetent individuals typically do not show symptoms after HCMV infections and the virus resides latently in the body throughout life. Under conditions of immunosuppression, as observed during human immunodeficiency virus infection or in transplant patients, viral reactivation can cause severe disease[1]. Furthermore, vertical transmission of HCMV can induce birth defects in newborns. In industrialized countries congenital HCMV infection currently is the leading cause of nonhereditary sensorineural impairments such as hearing loss[2]. Despite great efforts, no effective HCMV vaccine is licensed and currently available treatment regimens can limit virus replication, but do not eliminate latent viral reservoirs.

Both the innate and adaptive immune responses contribute to the control of HCMV infection[3]. Type 1 interferons (IFN-I) are important antiviral cytokines that are induced rapidly upon infection to promote and shape innate and adaptive immune responses[4]. Due to highly species-specific immune evasion mechanisms, human and mouse CMV show an obligate species tropism. Nevertheless, infection of mice with murine CMV (MCMV) recapitulates important characteristics of HCMV infection of humans, including the fact that the induction of IFN-I and triggering of its cognate IFN-I receptor (IFNAR) are essential to protect against lethal disease[5,6]. Following MCMV infection two waves of IFN-I responses are induced peaking between 4–8 h and 36–48 h post infection (hpi)[7].

Cells mount IFN-I responses upon triggering of pattern recognition receptors (PRR) by certain pathogen associated components[8–10]. Toll-like receptors (TLR) that signal via the adapter molecules myeloid differentiation primary response gene 88 (MyD88) and TIR-domain containing adaptor inducing IFN-β (TRIF) have previously been shown to be critically involved in triggering plasmacytoid dendritic cells (pDC) to mount IFN-I responses[11,12]. This mechanism is particularly important after MCMV infection for the induction of the second wave of IFN-I[13,14]. Correspondingly, MyD88 deficient mice have increased susceptibility to infection with MCMV isolated from salivary gland (SG) extracts[15]. Nevertheless, MyD88 deficient patients do not have an increased susceptibility to herpesvirus infections, implying redundancy of different PRR sensing mechanisms[16,17].

RIG-I-like receptors (RLR) recognize cytoplasmic RNA, particularly RNA viruses[8]. Identification of the cytoplasmic DNA receptor cyclic guanosine monophosphate–adenosine monophosphate synthase (cGAS) and its adapter molecule stimulator of IFN genes (STING) brought cytoplasmic DNA sensing of self and non-self DNA to the fore[18,19]. Upon binding of DNA cGAS catalyzes the formation of the secondary messenger 2′,3′-cyclic guanosine monophosphate–adenosine monophosphate (cGAMP) that binds STING and subsequently activates an antiviral cytokine response, including IFN-I[20,21]. While upon infection with DNA-encoded alpha-herpesvirus and gamma-herpesvirus, such as herpes simplex virus 1 (HSV-1)[22], varicella zoster virus[23], and murine gamma-herpesvirus 68[24] cGAS/STING signaling was needed to limit virus replication and to promote protection, the role of STING during the pathogenesis of CMV infection is less well understood. In vitro studies showed earlier that after HCMV infection of human monocyte-derived dendritic cells (DC) and macrophages[25] or murine bone marrow-derived DC (BMDC)[22] cGAS/STING signaling was essential to induce protective IFN-I responses. Furthermore, upon MCMV infection of BMDC or bone marrow-derived macrophages IFN-I responses are induced in a TLR/RLR-independent and STING-dependent manner[12,26]. Recently, it was suggested that STING signaling

contributes to the early systemic and splenic IFN-β production after MCMV infection[27,28]. However, how STING signaling influences the pathogenesis of MCMV in vivo still remains elusive.

Upon systemic MCMV infection, the virus rapidly infects cells in spleen and liver[29,30]. In a next step, especially blood mononuclear phagocytes are recruited to primary infection sites by the MCMV-encoded viral chemokine homolog MCK2. These cells then get infected, migrate to other organs and thus contribute to viral dissemination[31]. MCK2 facilitates the recruitment of monocytes[32,33], modulates antiviral CD8+ T cell immunity[34], and promotes the infection of macrophages[35]. Interestingly, upon infection of mice with a MCMV mutant devoid of MCK2 only endothelial cells, but not hepatocytes, support MCMV dissemination from the liver to other organs[36]. Finally, in the human system myeloid cells may get latently HCMV infected[37,38], whereas the role of myeloid cells in viral dissemination is being discussed controversially[37,39,40]. Whether STING signaling affects the dissemination of MCMV by myeloid cells and prevents viral spread from the liver to other organs has not been addressed, yet.

Here we perform a spatiotemporal analysis of STING vs. TLR/RLR signaling mediating the induction of IFN-β responses during MCMV infection. These experiments show that the three sensing platforms, TLR, RLR, and cGAS/STING, concomitantly contribute to the induction of protective anti-MCMV immune responses. Mice that lack single signaling platforms control the virus normally, whereas mice lacking all three ones are not able to mount IFN-I responses after MCMV infection and die within 6 days. Although, early IFN-I responses are induced in a STING-dependent manner, TLR/RLR signaling triggers IFN-I during the second wave. Interestingly, CD169+ Kupffer cells require STING signaling for the induction of early hepatic IFN-β responses and mice devoid of STING show increased virus titers in the liver at early time points. Importantly, the lack of STING enhances viral dissemination via myeloid cells to the lymph nodes, whereas it does not relieve the restriction of MCMV dissemination from hepatocytes to other organs.

## Results

**Combined TLR/RLR/STING signaling controls MCMV infection.** TLR signaling plays an important role in the control of systemic MCMV infection[15], whereas more recently it was discovered that STING signaling is needed to induce early systemic and splenic IFN-β responses[27,28]. To analyze the relevance of the cGAS/STING axis for protection against MCMV, TLR (*Myd88−/−Trif−/−* mice), RLR (*Mavs−/−* mice), cGAS (*Cgas−/−* mice), and STING (*Tmem173−/−* mice) signaling deficient mice were infected with $5 \times 10^5$ plaque forming units (pfu) of MCMV Δm157. All analyzed single signaling platform deficient mice showed signs of moderate disease and survived the infection similarly well as C57BL/6 (WT) control mice (Fig. 1a, b, Supplementary Fig. 1a, b). This observation was surprising, because in a previous in vivo study STING was reported to be involved in MCMV sensing and IFN-I induction[27,28] and mice devoid of cGAS or STING showed enhanced sensitivity to the infection with other DNA viruses, such as HSV-1 and vaccinia virus (VACV)[22,24] (Supplementary Fig. 1c, d). Mice with a combined TLR and RLR signaling deficiency (*Myd88−/−Trif−/−Mavs−/−*, here called MyTrMa knockout (KO) mice) showed enhanced vulnerability to MCMV infection and 60% of the animals died between day 6 and 8 (Fig. 1a, b). Interestingly, mice with a combined deficiency of TLR, RLR and cGAS (*Myd88−/−Trif−/−Mavs−/−Cgas−/−*, MyTrMaGa KO mice) or STING (*Myd88−/−Trif−/−Mavs−/−Tmem173−/−*, MyTrMaSt KO mice) signaling succumbed to the infection within 6 days (Fig. 1a–d) and therefore showed a similar sensitivity to lethal

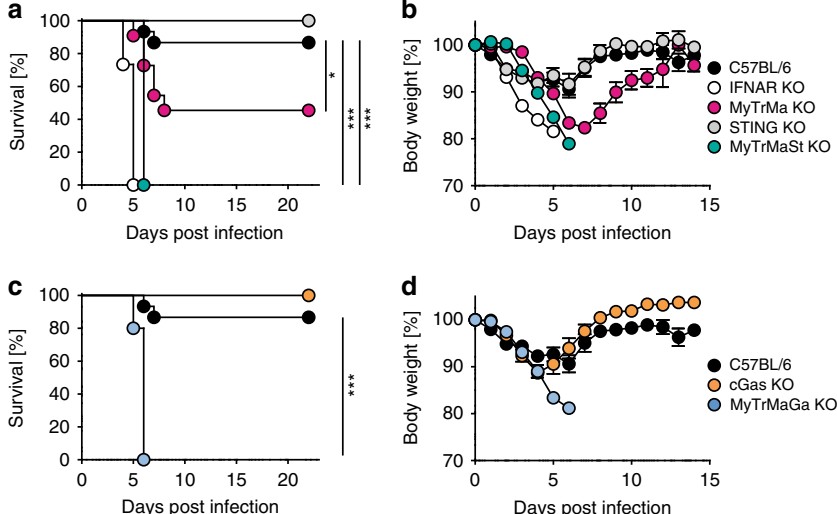

**Fig. 1** Combined TLR/RLR/STING signaling is needed to protect mice from lethal MCMV infection. C57BL/6 (n = 15), IFNAR KO (Ifnar$^{-/-}$, n = 15), MyTrMa KO (Myd88$^{-/-}$Trif$^{-/-}$Mavs$^{-/-}$, n = 11), STING KO (Tmem173$^{-/-}$, n = 9), and MyTrMaSt KO (Myd88$^{-/-}$Trif$^{-/-}$Mavs$^{-/-}$Tmem173$^{-/-}$, n = 9) mice were i.v. infected with 5 × 10$^5$ pfu MCMV Δm157 and **a** survival, as well as **b** body weight was monitored daily. Similarly, C57BL/6 (n = 15), cGAS KO (Cgas$^{-/-}$, n = 6), and MyTrMaGa KO (Myd88$^{-/-}$Trif$^{-/-}$Mavs$^{-/-}$Cgas$^{-/-}$, n = 5) mice were infected and **c** survival and **d** body weight were monitored daily. Survival and body weight of cGAS KO and MyTrMaGa KO mice were analyzed in the same experiments as the mice from Fig. 1a, b. Therefore, the depicted C57BL/6 control data are the same ones as in Fig. 1a, b. In case body weight decreased by more than 20% of the initial value, or when the overall health status was dramatically impaired, mice were sacrificed. Data represent at least two independently performed experiments. Error bars indicate mean ± s.e.m. (*p ≤ 0.0247, ***p ≤ 0.0001; a Log-rank (Mantel Cox) Test was used to calculate p-values)

MCMV infection as detected in IFNAR KO (Ifnar$^{-/-}$) mice (Fig. 1a, b). Thus, ablation of the cGAS/STING axis did not impair the control of MCMV infection, because TLR/RLR signaling alone was sufficient to induce protective anti-viral immunity. Nevertheless, the experiments with TLR/RLR/STING or TLR/RLR/cGAS signaling deficient mice revealed that the cGAS/STING axis contributed to the survival of approximately 40% of infected mice devoid of TLR/RLR signaling. These experiments highlight the in vivo relevance of the cGAS/STING axis during MCMV infection.

**MCMV sensing by STING induces early IFN-I serum responses.** As previously shown by others[6] and confirmed by us, IFNAR KO mice succumb to MCMV infection within 5 days (Fig. 1a) underscoring the relevance of IFNAR triggering for the protection against MCMV infection. Upon MCMV infection IFN-I responses are induced in two waves. While it is not entirely clear which signaling platforms elicit the first wave, the second wave is conferred primarily by pDC in a TLR-dependent manner[13,14]. The analysis of serum IFN-I at 4 hpi revealed that mice in which only the cGAS/STING axis remained intact while TLR/RLR signaling was impaired (MyTrMa KO) showed moderately reduced IFN-α and enhanced IFN-β responses when compared with WT mice (Fig. 2a, b). Interestingly, in the serum of infected STING KO mice early IFN-α and IFN-β responses were significantly reduced (Fig. 2a, b). In contrast, at 36 hpi MyTrMa KO mice neither showed IFN-α nor IFN-β responses, whereas in infected STING KO mice enhanced and normal IFN-α and IFN-β levels were detected, respectively (Fig. 2a, b). Importantly, in infected MyTrMaSt KO mice no serum IFN-I responses were detected at any of the time points tested (Fig. 2a, b). Interestingly, the STING-dependent induction of early serum IFN-β was detected in infection experiments with m157 deficient, as well as m157 proficient MCMV (Supplementary Fig. 2). These data indicate that STING signaling plays an important role for the induction of early IFN-I, whereas the second IFN-I wave is dependent on TLR/RLR signaling and independent of the cGAS/STING axis.

**MCMV induces early hepatic IFN-β in a STING-dependent manner.** To study the role of the different signaling platforms in IFN-β induction during MCMV infection in an organ-specific manner, we intercrossed IFN-β reporter mice (Ifnb$^{wt/}$ $^{Δβluc}$ mice)[41] with the different PRR signaling deficient mice. In accordance with the above serum data, infected IFN-β reporter mice with intact signaling platforms showed strong bioluminescence imaging (BLI) signals already at 4 hpi on the ventral side of the abdominal region (Fig. 3a and Supplementary Fig. 3a). At 24 hpi a second signal detected on the left side of the mouse appeared that peaked at 48 hpi, and that was followed by signals in the cervical and inguinal area from 48 hpi on until 5 days post infection (dpi) (Fig. 3a and Supplementary Fig. 3a). Ex vivo luminescence analysis of isolated liver and spleen revealed the organ origin of the BLI signals identified above (Fig. 3c–e). At 4 hpi STING competent MyTrMa KO IFN-β reporter mice (Myd88$^{-/-}$Trif$^{-/-}$Mavs$^{-/-}$Ifnb$^{wt/Δβluc}$ mice) showed enhanced BLI signals in the liver when compared with IFN-β reporter mice, while in STING KO IFN-β reporter mice (Tmem173$^{-/-}$Ifnb$^{wt/}$ $^{Δβluc}$ mice) hepatic BLI signals were absent (Fig. 3a–e). In contrast, at 36 hpi BLI signals were significantly reduced in the spleen of MyTrMa KO IFN-β reporter mice, whereas in STING KO IFN-β reporter mice similar BLI signals were detected as in IFN-β reporter controls. In MyTrMa KO IFN-β reporter mice the splenic BLI signal increased between 36 and 48 hpi, nevertheless, compared with IFN-β reporter controls the signal intensity was still reduced at 48 hpi. In contrast, at that time STING KO IFN-β reporter mice showed similar splenic BLI signals as IFN-β reporter controls (Fig. 3a–e). Of note, the STING-mediated IFN-β induction was independent of the presence of TRIF (Supplementary Fig. 3b). Although in a previous study STING signaling was shown to be TRIF-dependent following in vitro stimulation with synthetic STING ligands[42], under in vivo conditions TRIF was not needed to mount STING-dependent IFN-β responses after MCMV infection. Finally, the first wave of IFN-β induction in the liver is driven in a STING-dependent manner, whereas

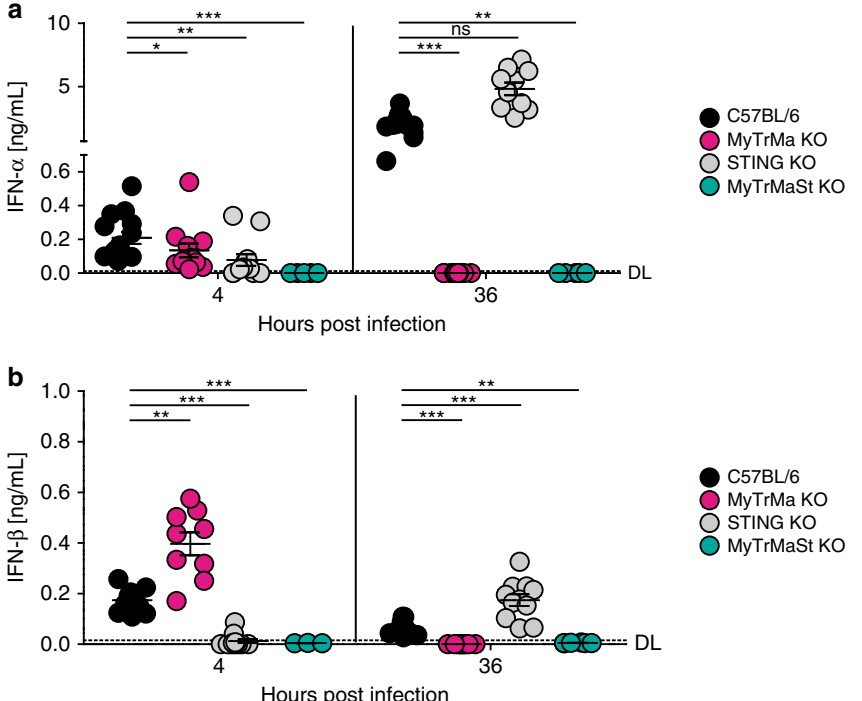

**Fig. 2** STING signaling induces the first wave of IFN-I responses after MCMV infection. C57BL/6, MyTrMa KO (Myd88$^{-/-}$Trif$^{-/-}$Mavs$^{-/-}$), STING KO (Tmem173$^{-/-}$), and MyTrMaSt KO (Myd88$^{-/-}$Trif$^{-/-}$Mavs$^{-/-}$Tmem173$^{-/-}$) mice were i.v. infected with 5 × 10$^5$ pfu MCMV Δm157. Blood was drawn at 4 and 36 hpi and serum was analyzed for **a** IFN-α levels by an ELISA method. Data are from at least two independent experiments. (C57BL/6 n = 14 (4 hpi), 13 (36 hpi); MyTrMa KO n = 12, STING KO n = 12 (4 hpi), 10 (36 hpi), MyTrMaSt KO n = 5 (4 hpi), 6 (36 hpi); *p ≤ 0.0253, **p ≤ 0.0018, ***p ≤ 0.0006; ns = not statistically significant; a two-tailed Mann–Whitney test was used to calculate p-values). Blood was drawn at 4 and 36 hpi and analyzed for **b** IFN-β levels by an ELISA method. Data are from at least two independent experiments. (C57BL/6 n = 11, MyTrMa KO n = 9, STING KO n = 12 (4 hpi), 11 (36 hpi); MyTrMaSt KO n = 4 (4 hpi), 6 (36 hpi); **p ≤ 0.005, ***p ≤ 0.0008; a two-tailed Mann–Whitney test was used to calculate p-values). Dashed line (DL = Detection Limit). Error bars indicate mean ± s.e.m.

TLR/RLR signaling is needed for the induction of the second IFN-β wave in the spleen.

**MCMV triggers Kupffer cells to mount early IFN-β responses.** To specify which liver resident cell subset mounts IFN-β responses after MCMV infection, conditional IFN-β reporter mice were analyzed. Infection experiments with Alb$^+$IFN-β reporter mice (AlbCre$^{+/-}$Ifnb$^{flox-βluc}$ mice), carrying a functional reporter only in Albumin$^+$ (Alb$^+$) hepatocytes, revealed that hepatocytes did not contribute to the IFN-β induction in the liver between 4 and 96 hpi (Fig. 4a, b). CD11c$^+$IFN-β reporter mice (ItgaxCre$^{+/-}$Ifnb$^{flox-βluc}$ mice) that carry a functional reporter only in CD11c$^+$ cells, such as DC and certain macrophage subsets, showed minor IFN-β expression at 4 hpi in the liver. In contrast, experiments with LysM$^+$IFN-β reporter mice (Lyz2Cre$^{+/-}$Ifnb$^{flox-βluc}$ mice) and CD169$^+$IFN-β reporter mice (CD169Cre$^{+/-}$Ifnb$^{flox-βluc}$ mice) revealed that Lysozyme M$^+$ (LysM$^+$) and CD169$^+$ cells, i.e., LysM$^+$/CD169$^+$ Kupffer cells (KC), significantly contributed to hepatic IFN-β responses (Fig. 4a, b). Interestingly, analysis of the spleen showed that the IFN-β induction between 24 and 36 hpi was predominantly mediated by CD11c$^+$ cells, and to a lesser extent by LysM$^+$ or CD169$^+$ cells (Fig. 4c, d). However, at 48 hpi also LysM$^+$ and CD169$^+$ cells showed enhanced IFN-β expression in the spleen (Fig. 4c, d). Taken together, upon MCMV infection CD169$^+$ KC mount early IFN-β responses in the liver, whereas mainly CD11c$^+$ cells show IFN-β expression during the second IFN-β wave in the spleen. At later time points, also LysM$^+$ and CD169$^+$ myeloid cells contribute to splenic IFN-β expression.

**STING signaling controls early hepatic MCMV.** To address whether differences in local IFN-β induction affect MCMV replication and dissemination, mice were intravenously (i.v.) infected with a luciferase expressing MCMV Δm157[43] and virus replication and dissemination was studied by in vivo imaging. As expected, the first target organs showing BLI signals after infection were the liver and spleen (Fig. 5a). At 1.5–3 dpi moderate MCMV infection was also detected in the inguinal and the cervical area, whereas at 8 dpi the virus was mainly present in the cervical area (Fig. 5a). While IFNAR KO mice could not control virus replication and BLI signals were detected in the whole body by 3 dpi, MyTrMa KO mice showed enhanced virus replication in the liver, the inguinal, and the cervical area between 3 and 8 dpi (Fig. 5a–c). Importantly, MyTrMaSt KO mice showed the highest virus replication already at 1.5 dpi, which was not controlled and led to death of the animals by 6 dpi, highlighting the significance of concomitant MyTrMa and STING signaling for the control of MCMV infection. Surprisingly, liver BLI signals in STING KO and WT mice were similar between 1.5 and 8 dpi, whereas STING deficiency caused a locally enhanced signal in the cervical area at 8 dpi (Fig. 5a–c). MCMV titer analysis of organ homogenates revealed a significant contribution of STING signaling to the control of the initial hepatic virus burden (Fig. 5d). Nevertheless, over time STING KO mice were able to control virus replication in the liver. This is in contrast to MyTrMa deficient mice that showed increased virus titers between 1 and 5 dpi (Fig. 5d). The analysis of virus titers in MyTrMa KO and MyTrMaSt KO mice also revealed a contribution of STING signaling to the initial control of MCMV in the spleen, while at later time points mainly MyTrMa signaling restricted MCMV replication (Fig. 5e). Thus,

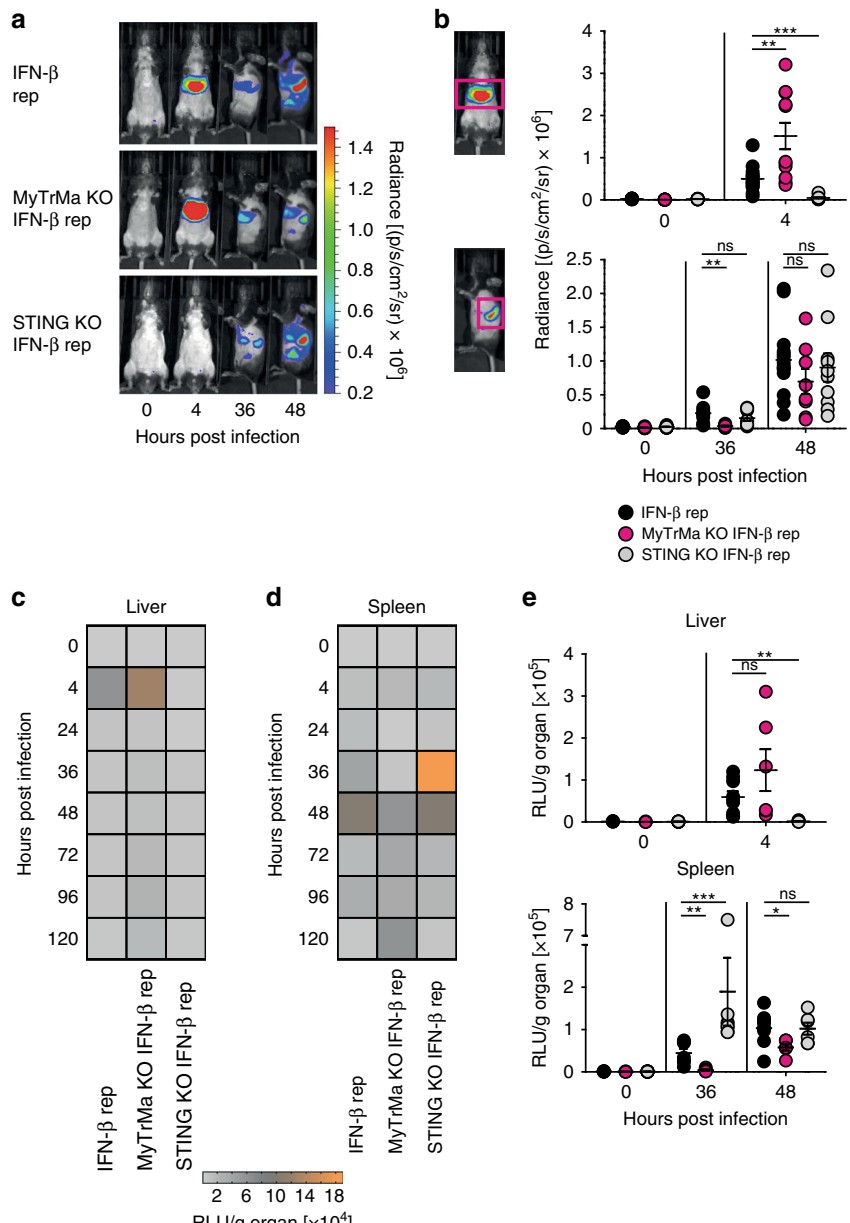

**Fig. 3** Early hepatic IFN-β is induced in a STING-dependent manner. IFN-β reporter (rep) (*Ifnb^wt/Δβ-luc*), MyTrMa KO IFN-β reporter (*Myd88^−/−Trif^−/−Mavs^−/−Ifnb^wt/Δβ-luc*), and STING KO IFN-β reporter (*Tmem173^−/−Ifnb^wt/Δβ-luc*) mice were i.v. infected with $5 \times 10^5$ pfu MCMV Δm157. **a** At the indicated time points luciferin was i.v. injected and luciferase activity was monitored by in vivo imaging. **b** Ventral and left flank regions were marked as regions of interest and bioluminescence imaging signals were quantified. Data represent at least two independently performed experiments. One representative mouse out of at least seven similar ones is shown. (IFN-β reporter $n \geq 8$, MyTrMa KO IFN-β reporter $n \geq 7$, STING KO IFN-β reporter $n \geq 8$, **$p \leq 0.0067$, ***$p < 0.0001$; ns = not statistically significant; a two-tailed Mann-Whitney test was used to calculate *p*-values). Luminescence was measured in relative light units (RLU) of **c** liver and **d** spleen homogenates ex vivo. Mean values are shown. **e** Ex vivo measured luminescence of liver and spleen homogenates was quantified. Data represent at least two independently performed experiments. (IFN-β reporter $n \geq 6$, MyTrMa KO IFN-β reporter $n \geq 5$, STING KO IFN-β reporter $n \geq 6$; *$p \leq 0.0452$, **$p \leq 0.0025$, ***$p \leq 0.0003$; ns = not statistically significant; a two-tailed Mann-Whitney test was used to calculate *p*-values). Error bars indicate mean ± s.e.m.

STING signaling is important for the initial control of MCMV not only in the liver but also in the spleen. At later stages of the infection, STING also limits the viral burden in the cervical area. In contrast, MyTrMa signaling is crucial for the limitation of MCMV replication after the initial phase of the infection.

**STING is needed to restrict MCMV dissemination to lymph nodes**. A previous study showed that hepatocyte-derived MCMV did not further disseminate to other organs[36]. In this study a

MCK2-negative floxed-stop reporter MCMV (MCMV Δm157-flox-*egfp*) was used that showed onset of eGFP expression after passage through Cre expressing cells (MCMVrec) (Fig. 6a). Interestingly, endothelial cell derived MCMV disseminated to various tissues, including spleen, lung, and others, whereas in *AlbCre^+/−* mice hepatocyte derived MCMVrec was not detected in any of these organs until 5 dpi (Fig. 6b)[36]. Myeloid cells were discussed as being of key relevance for MCMV dissemination[32,33]. Furthermore, in a previous study we found that after CMV infection STING is especially important in

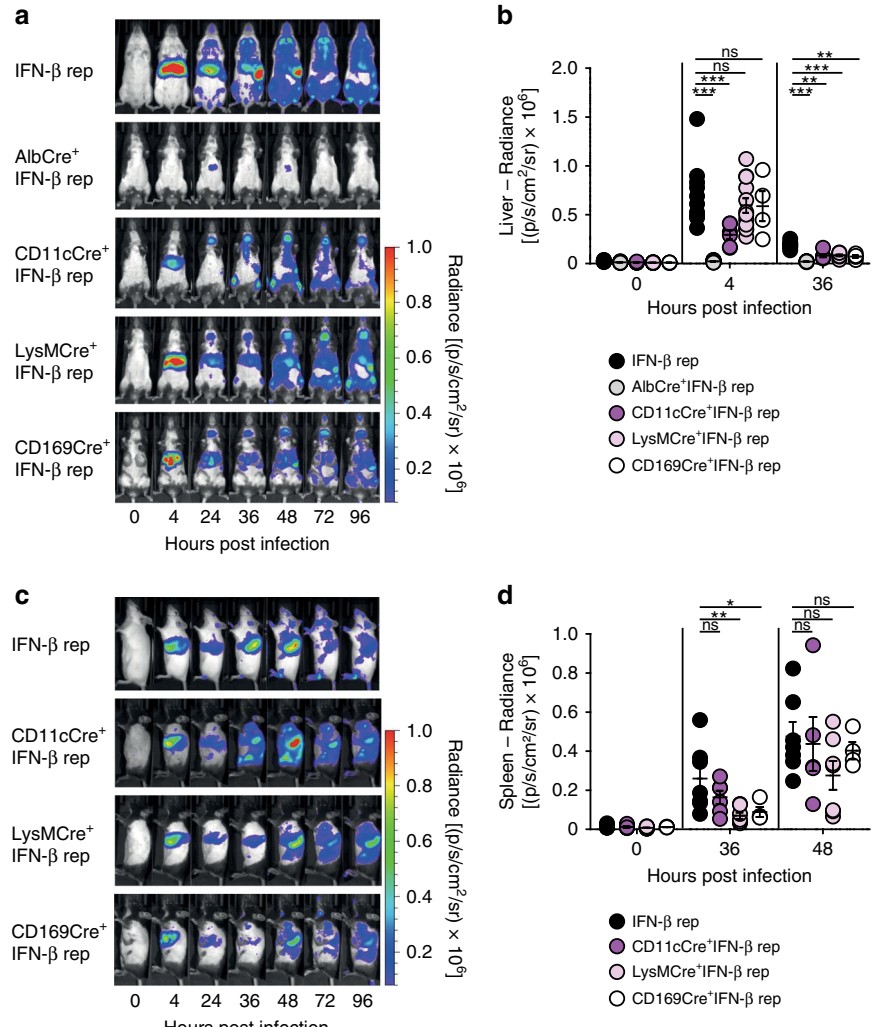

**Fig. 4** Liver-resident Kupffer cells mount IFN-β responses in a STING-dependent manner. IFN-β reporter (rep) (*Ifnb*^wt/Δβ-luc^), AlbCre⁺IFN-β reporter (*AlbCre*^+/−^*Ifnb*^floxβ-luc^), CD11cCre⁺IFN-β reporter (*ItgaxCre*^+/−^*Ifnb*^floxβ-luc^), LysMCre⁺IFN-β reporter (*Lyz2Cre*^+/−^*Ifnb*^floxβ-luc^), and CD169Cre⁺IFN-β reporter (*CD169Cre*^+/−^*Ifnb*^flox-βluc^) mice were i.v. infected with 5 × 10⁵ pfu MCMV Δm157. **a** At the indicated time points luciferin was i.v. injected and luciferase activity was monitored by in vivo imaging. **b** Liver was marked as region of interest and luminescence intensity was quantified. **c** At the indicated time points luciferin was injected i.v. and luciferase activity was monitored by in vivo imaging. **d** Spleen was marked as region of interest and luminescence intensity was quantified. Data represent at least two independently performed experiments and one representative mouse out of at least four similar ones is shown. Error bars indicate mean ± s.e.m. (IFN-β reporter *n* ≥ 7, AlbCre⁺IFN-β reporter *n* ≥ 9, CD11cCre⁺IFN-β reporter *n* ≥ 5, LysMCre⁺IFN-β reporter *n* ≥ 7, CD169Cre⁺IFN-β reporter *n* = 4; *p ≤ 0.0424, **p ≤ 0.0061, ***p ≤ 0.0006; ns = not statistically significant; a two-tailed Mann–Whitney test was used to calculate *p*-values)

myeloid cells for the induction of IFN-I responses[25]. To address whether STING signaling is important in myeloid cells to control MCMV dissemination, we generated a repaired version of the reporter virus that contains a wildtype MCK2 locus (MCMVrep Δm157-flox-egfp). Infection of myeloid cell-specific Cre mice either on a STING proficient (*Lyz2Cre*^+/−^*Tmem173*^wt/wt^, LysMCre⁺STING WT mice) or STING deficient (*Lyz2Cre*^+/−^*Tmem173*^−/−^, LysMCre⁺STING KO mice) background with the repaired reporter MCMV revealed enhanced MCMV titers at 3 dpi in the liver of LysMCre⁺STING KO mice when compared with LysMCre⁺STING WT mice (Fig. 6c and Supplementary Fig. 4a). In accordance with the important role of STING signaling in KC for the induction of IFN-β responses, higher numbers of virus particles derived from productively infected myeloid cells were detected in the liver of STING deficient than of WT mice. In all other analyzed organs MCMV titers were similar in WT and STING deficient mice and similar

numbers of MCMVrec were detected in spleen, inguinal lymph nodes (iLN), and cervical lymph nodes (cLN) (Fig. 6c). Surprisingly, in contrast to 3 dpi, at 8 dpi particularly in the iLN and cLN higher myeloid cell derived MCMVrec levels were detected in LysMCre⁺STING KO mice, which was also manifested on the level of enhanced overall MCMV titers (Fig. 6c and Supplementary Fig. 4c). No differences in MCMV titers or MCMVrec levels were detected in LysMCre⁺STING WT and LysMCre⁺STING KO mice in the other analyzed organs (Fig. 6c).

To further analyze whether STING is essential to restrict the dissemination of MCMV from hepatocytes to other organs, we infected AlbCre⁺STING WT (*AlbCre*^+/−^*Tmem173*^wt/wt^ mice) and AlbCre⁺STING KO (*AlbCre*^+/−^*Tmem173*^−/−^ mice) mice with the reporter MCMV. At 3 dpi MCMVrec was detected only in the liver of AlbCre⁺STING WT and AlbCre⁺STING KO mice and not in other organs (Fig. 6d), which was similarly reported for the MCK2-negative MCMV variant before[36]. Interestingly, at 8 dpi

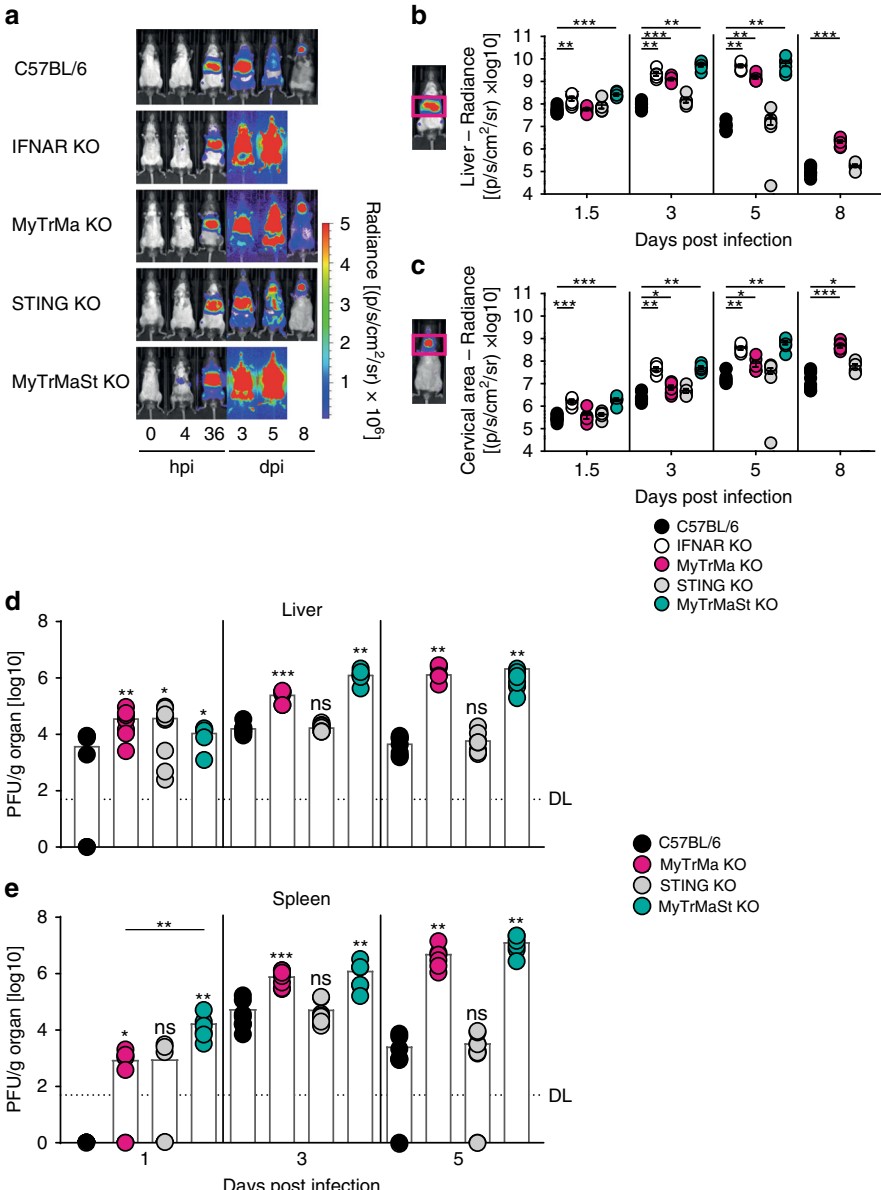

**Fig. 5** STING is important for the control of early hepatic MCMV. C57BL/6, IFNAR KO (*Ifnar*−/−), MyTrMa KO (*Myd88*−/−*Trif*−/−*Mavs*−/−), STING KO (*Tmem173*−/−), and MyTrMaSt KO (*Myd88*−/−*Trif*−/−*Mavs*−/−*Tmem173*−/−) mice were i.v. infected with $5 \times 10^5$ pfu MCMV Δ*m157luc*. **a** At the indicated time points luciferin was i.v. injected and the luciferase activity was monitored by in vivo imaging. One representative mouse out of six similar ones is shown. **b** Liver and **c** the cervical area were marked as region of interest and luminescence signals were quantified. Data are from at least two independently performed experiments. Error bars indicate mean ± s.e.m. (C57BL/6 $n \geq 7$, IFNAR KO $n = 6$, MyTrMa KO $n \geq 6$, STING KO $n = 6$, MyTrMaSt KO $n = 6$; *$p \leq 0.0350$, **$p \leq 0.0047$, ***$p \leq 0.0007$; a two-tailed Mann-Whitney test was used to calculate *p*-values). C57BL/6, MyTrMa KO, STING KO, and MyTrMaSt KO mice were infected as indicated above with MCMV Δ*m157* and mice were perfused with PBS on 1, 3, and 5 dpi. **d** Liver and **e** spleen were removed, weighed and viral titers in pfu per gram organ were determined by a plaque assay. Data are from at least two independently performed experiments. Dashed line (DL = Detection Limit). Bars indicate mean (C57BL/6 $n = 7$, MyTrMa KO $n \geq 6$, STING KO $n \geq 8$, MyTrMaSt KO $n = 6$; *$p \leq 0.0381$, **$p \leq 0.0072$, ***$p \leq 0.0006$; ns = not statistically significant; a two-tailed Mann–Whitney test was used to calculate *p*-values)

MCMVrec was additionally detected in the SG of both mouse strains, but not in other analyzed organs, and the titers of MCMVrec were very similar in WT and STING deficient AlbCre+ mice (Fig. 6d).

Taken together, STING signaling is important in myeloid cells to constrain MCMV infection and to limit myeloid cell-mediated dissemination of MCMV, whereas until day 3 pi STING signaling does not play a role in restricting viral dissemination from hepatocytes to other organs, and it does not affect MCMV dissemination from hepatocytes to the SG between day 3 and 8.

## Discussion

Understanding the pathogenesis of cytomegalovirus infection is imperative to improve preventive and symptomatic treatments. Here we carried out a spatiotemporal analysis of the role of cGAS/STING in the induction of IFN-β responses during MCMV infection and in MCMV dissemination. Surprisingly, cGAS/STING signaling alone was not needed to promote survival after MCMV infection. However, concomitant TLR, RLR, and cGAS/STING signaling was necessary to protect mice from lethal MCMV infection. The first wave of the protective biphasic IFN-I

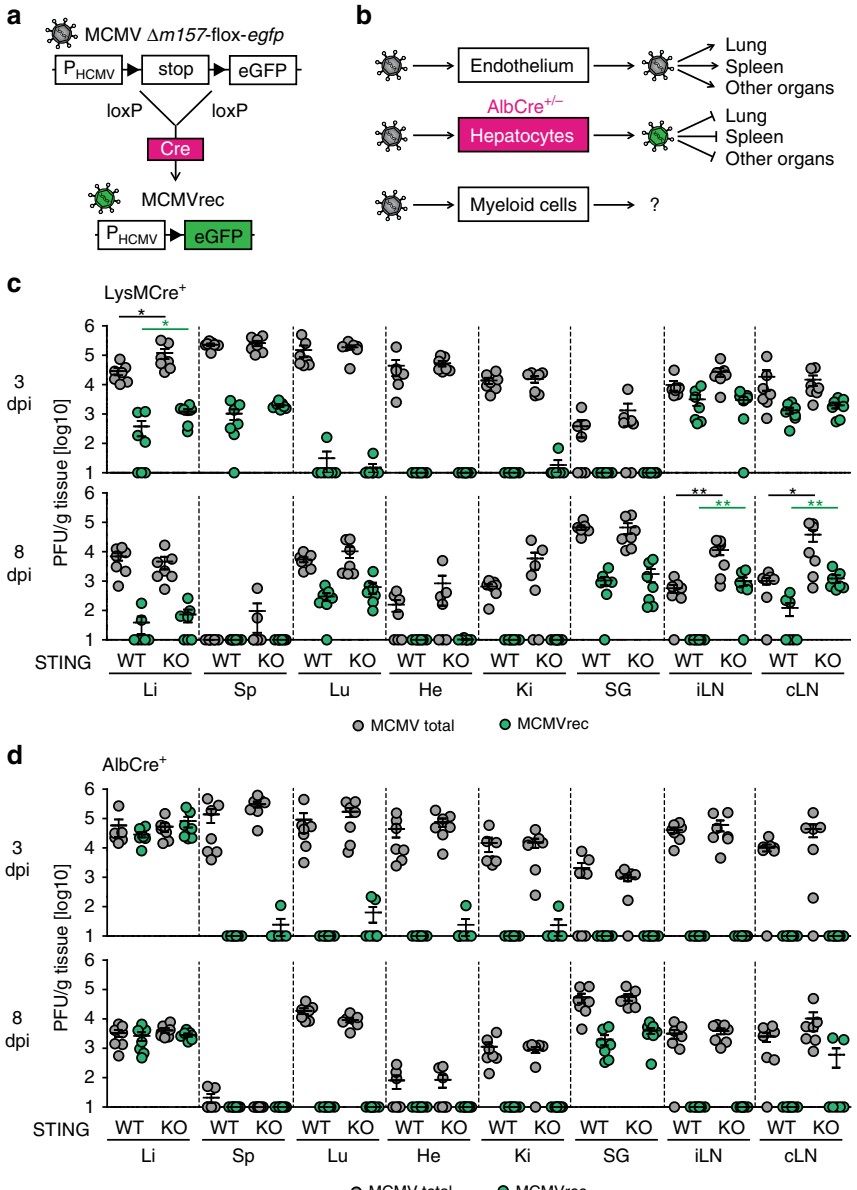

**Fig. 6** STING signaling limits MCMV replication in myeloid cells and restricts viral dissemination to LN. Schematic depiction of the viral reporter system that allows quantification of viral dissemination: **a** Upon growth of the reporter virus MCMV Δ*m157*-flox-*egfp* in Cre expressing cells the loxP flanked STOP is removed from the viral genome and eGFP expression is constitutively induced. **b** Hepatocyte-derived MCMV does not infect other peripheral organs[36]. **c–d** STING proficient and STING deficient mice expressing Cre either in myeloid cells (*Lyz2Cre*[+/−]*Tmem173*[wt/wt], LysMCre[+]STING WT and *Lyz2Cre*[+/−]*Tmem173*[−/−], LysMCre[+]STING KO mice, respectively) or in hepatocytes (*AlbCre*[+/−]*Tmem173*[wt/wt], AlbCre[+]STING WT and *AlbCre*[+/−] *Tmem173*[−/−], AlbCre[+]STING KO mice, respectively) were i.v. infected with 5 × 10[5] pfu MCMVrep Δ*m157*-flox-*egfp* and perfused at 3 and 8 dpi. Liver (Li), spleen (Sp), lung (Lu), heart (He), kidney (Ki), salivary glands (SG), inguinal lymph nodes (iLN), and cervical lymph nodes (cLN) were prepared and virus titers in pfu per g organ were analyzed by a plaque assay. GFP[−] total MCMV plaques (gray circles) and GFP[+] MCMVrec plaques (green circles) of **c** LysMCre[+]STING WT and LysMCre[+]STING KO mice or **d** AlbCre[+]STING WT and AlbCre[+]STING KO mice were counted using either a light microscope or a fluorescence microscope. Data represent at least two independently performed experiments. Error bars indicate mean ± s.e.m. (*n* = 7 for all analyzed genotypes; \**p* ≤ 0.0262, \*\**p* ≤ 0.0064; a two-tailed Mann–Whitney test was used to calculate *p*-values)

induction was STING-dependent, while TLR/RLR signaling was essential for the second wave. Specifically, liver-resident KC showed a STING-dependent IFN-β induction and correspondingly in the liver of infected STING KO mice enhanced MCMV titers were detected. Nevertheless, STING did not control the dissemination of MCMV from hepatocytes to other organs, whereas it was needed to control virus dissemination via myeloid cells to lymph nodes.

We found that cGAS KO and STING KO mice did not show enhanced susceptibility to lethal MCMV infection, whereas

MyTrMa KO mice that are devoid of TLR and RLR signaling showed enhanced vulnerability to lethal MCMV infection. These data were in accordance with an earlier report indicating that *Myd88*[−/−] mice did not control MCMV infection as efficiently as WT controls[15]. However, unlike Delale et al. we did not observe a major difference in survival of MyTr KO and WT mice, which presumably was due to the use of salivary gland extract isolated MCMV in the study by Delale et al. vs. cell culture derived MCMV in our experiments. Nevertheless, our data clearly support the hypothesis that defects in the cGAS/STING axis are

compensated by TLR/RLR signaling. Interestingly, cGAS KO and STING KO mice showed increased virus loads and enhanced disease after infection with a whole variety of DNA encoded viruses, including poxviruses such as VACV (Supplementary Fig. 1c, d)[24] and ectromelia virus[44], as well as other herpesviruses such as HSV-1[22]. Unlike MCMV, VACV and ectromelia virus replicate in the cytoplasm. Thus, in these cases the viral genome is readily accessible to sensing by cGAS. Because both HSV-1 and MCMV are sensed in macrophages in a cGAS/STING-dependent manner[22,26], it is surprising that mice deficient of the cGAS/STING axis showed enhanced sensitivity to HSV-1 infection[22], whereas they were as resistant to MCMV infection as WT mice. Why the cGAS/STING axis has such divergent roles in two different herpesvirus infections is currently difficult to understand and has to be addressed in future studies.

In TLR/RLR deficient (MyTrMa KO) mice the additional deletion of STING or cGAS (MyTrMaSt KO or MyTrMaGa KO) dramatically enhanced the susceptibility to lethal MCMV infection (from 60% to 100% lethality) and mice of both genotypes succumbed to the infection with similar kinetics as observed in IFNAR KO mice. These data verify that concomitant signaling of TLR, RLR, and cGAS/STING elicits protective immunity to MCMV. Moreover, despite the deletion of cGAS or STING does not confer a dramatically enhanced disease phenotype after MCMV infection, the cGAS/STING axis functions as an important sensor of MCMV in vivo and promotes survival of approximately 40% of the mice, as indicated by the comparative infection studies with TLR/RLR and TLR/RLR/STING or TLR/RLR/cGAS signaling deficient mice.

The spatiotemporal analysis of the contribution of STING signaling to the induction of IFN-α/β responses revealed that STING signaling was needed to induce early hepatic IFN-β responses. These early responses were mediated by LysM+ CD169+ KC, and not by Alb+ hepatocytes. Therefore, we concluded that KC sense MCMV via the cGAS/STING axis and mount IFN-β responses in a STING-dependent manner. This finding was surprising because we and others showed that hepatocytes are major targets of MCMV infection in the liver[30,36], and upon infection with other viruses, such as the RNA-encoded enterovirus Coxsackie virus B3, hepatocytes are the major IFN-β producers[45]. It is well possible that MCMV infected hepatocytes do not mount IFN-β responses because they do not express STING[46]. Furthermore, MCMV presumably evades the antiviral immune response in hepatocytes by counteracting PRR signaling pathways, which additionally would prevent the induction of IFN-β in hepatocytes. However, this possibility has to be addressed in future studies. A previous study described that upon systemic MCMV infection endothelial cells within the liver showed even higher initial infection rates than KC and hepatocytes[30]. Due to their localization at the liver sinusoids, KC and endothelial cells form the first barrier for pathogens that enter the liver via the portal vein. The capability of endothelial cells to produce IFN-I following CMV infection has previously been shown in experiments with isolated liver sinusoidal endothelial cells[47] and with human umbilical vein endothelial cells, the latter of which showed a cGAS-STING-dependent induction of IFN-I responses[27]. To study IFN-β expression by KC we used CD169-specific IFN-β reporter mice. CD169 is a lectin-like receptor that is primarily expressed on tissue-resident macrophages and that is not found on CD31 positive endothelial cells[48,49]. Thus, by studying CD169-specific IFN-β reporter mice Cre-mediated targeting of endothelial cells could be excluded. Based on the above in vitro data it is likely that under in vivo conditions endothelial cells can contribute to early IFN-β expression, however, only to a very minor extent. Accordingly, we hypothesize that under in vivo conditions STING signaling plays an important role especially in KC to mount abundant early IFN-β responses, which reduce hepatic MCMV replication.

We found that during MCMV infection early IFN-β induction in the spleen is independent of signaling via either TLR, RLR, or STING. Previous studies reported that in the spleen initially fibroblastic reticular cells are infected by MCMV and that splenic IFN-I responses are induced in a TLR-independent manner[7,29]. In contrast, we found that IFN-β responses detected at 36 hpi are conferred by CD11c+ cell types in a MyTrMa-dependent manner. Others showed before that pDC, which express CD11c and sense MCMV in a TLR-dependent manner, produce IFN-I at 36 hpi, whereas CD11c+ conventional DC contribute to this IFN-I only to a very minor extent[13–15,50]. In addition, marginal zone, as well as red pulp macrophages were reported to co-express CD11c and CD169[49,51]. Therefore, these cells potentially contribute to the IFN-β induction at 36 hpi. However, at 48 hpi we found that additional sensing platforms are involved in the induction of IFN-β responses in the spleen, which can even partially compensate for the loss of MyTrMa signaling. These IFN-β responses seem to be contributed by a LysM+ and/or CD169+ cell subset in a STING-dependent manner. After systemic MCMV infection, cells in the marginal zone and the red pulp are initially infected, followed by the appearance of infected CD11c+ and CD11b+ cells, including conventional DC (LysM+), macrophages (LysM+ and CD169+) and granulocytes (LysM+) in the white pulp at 48 hpi[29,52]. CD11c+ pDC were shown to be barely productively infected by MCMV[12,53] or HCMV[25,54] in vitro. Therefore, we conclude that in the spleen the second wave of IFN-β is initially derived from pDC, whereas at later time points splenic macrophages contribute to IFN-β responses in a STING-dependent manner, when the contribution of pDC derived IFN-I declines.

The investigation of viral dissemination via LysM+ cells, including KC, revealed an enhanced productive infection of myeloid cells within the liver of STING deficient mice. This observation is consistent with the STING-dependent IFN-β induction in KC. IFNAR triggering leads to the induction of a whole variety of antiviral genes[4]. Thus, lack of IFN-I production by KC in STING deficient animals reduces the local antiviral state, which might render myeloid cells of the liver more susceptible to productive virus infection. Remarkably, at 8 dpi we detected significantly more myeloid-cell derived MCMVrec in the iLN and cLN of LysMCre+STING KO mice than in LysMCre+STING WT mice. This phenotype manifested also in enhanced virus titers in the iLN and cLN, as similarly observed in the whole cervical area of STING KO mice (Fig. 5c), whereas the overall virus titer difference was not so evident in AlbCre+ STING KO mice when compared with AlbCre+STING WT mice. In absence of STING, myeloid cells might be more permissive for MCMV infection and virus replication might be facilitated, which subsequently would lead to an increased transport of virus particles. The dissemination of MCMV relies on the transport by myeloid cells[31–33]. After intranasal infection, dissemination of MCMV from the lung to lymph nodes was reported to be mediated primarily by CD11c+ cell types[55]. However, innate immune responses that might influence the dissemination of MCMV via specific cell types have not been identified so far. In this context, we could show that STING signaling in myeloid cells is important for constraining infection, as well as viral dissemination.

Infection of hepatocytes by a MCMV variant deficient for the viral chemokine homolog MCK2 resulted in only very minor numbers of hepatocyte-derived virus particles detected in other target organs until 5 dpi[36]. We confirmed this observation and showed that also after infection with a MCK2-positive floxed-stop MCMV reporter virus MCMV does not disseminate from hepatocytes to other organs until 3 dpi. Nevertheless, at 8 dpi we

detected hepatocyte-derived MCMV in the SG, but not in other organs. Interestingly, the dissemination of MCMV from hepatocytes into the SG was unaltered when STING signaling was lacking. MCK2 is reported to recruit myeloid cells to sites of infection, where they get MCMV infected and are used as vehicles for viral dissemination[32,56]. In addition, MCK2 was described to promote the infection efficacy of macrophages[35]. In absence of MCK2, accumulation of patrolling monocytes was reduced and resulted in diminished dissemination to the SG[32]. Thus, in the classical experiments with the MCK2 deficient reporter virus mechanisms relevant for the dissemination from the hepatocytes to the SG might have been missed. Although STING is important for an efficient IFN-β induction by KC and the control of the initial virus load in the liver, absence of STING did not affect the restriction of dissemination from hepatocytes to other organs. Natural killer cells, T cells, and the endothelial barrier, as well as γ-irradiation sensitive immune cells were excluded as effector cells accounting for the restriction of virus dissemination from hepatocytes to other organs[36]. However, KC are not efficiently deleted from the liver by γ-irradiation, and IFN-I responses are well known to be important for the control of MCMV infection, leaving open of whether KC controlled viral dissemination. Nevertheless, with our system we proved that the restriction of MCMV dissemination from hepatocytes to other organs is independent of KC-derived IFN-β.

As similarly detected in immune cells of murine origin[13–15], also HCMV stimulated human pDC produce major amounts of IFN-I in a TLR-dependent manner[25,57], whereas monocyte-derived DC and macrophages or BMDC mount IFN-I responses in a STING-dependent manner[22,25]. Therefore, it is possible that also in the human system STING deficiency can be compensated by functional TLR signaling. Thus, it is very unlikely that single nucleotide polymorphisms that influence the functionality of components of the cGAS/STING axis affect the susceptibility of healthy individuals to HCMV infection. Nevertheless, our data indicate that the cGAS/STING axis alone was able to rescue a significant proportion of mice devoid of TLR and RLR signaling from lethal MCMV infection. Thus, in immunocompromised patients, in whom HCMV can cause major health problems, the functionality of STING might be of higher relevance. Interestingly, MyD88 deficient patients do not show enhanced susceptibility to herpesvirus infections, but under such conditions a functional cGAS/STING axis might be essential to compensate for the loss of TLR signaling[16,17]. Importantly, MCMV and HCMV were reported to establish latency in myeloid lineage cells[37,58,59]. Whether STING deficiency increases latent viral reservoirs and/or facilitates reactivation from latency is an important question, which remains to be addressed.

Herein, we showed that concomitant TLR/RLR/STING signaling is needed in order to induce an efficient innate immune response against MCMV. Although STING deletion alone has no effect on survival, STING is needed for the induction of early hepatic IFN-β responses, which are mediated primarily by KC and initially control MCMV replication within the liver. Furthermore, with the use of an improved reporter virus that allows tracking of MCMV replication in a given cell type, we showed that STING does not contribute to the restriction of MCMV dissemination from hepatocytes to other organs. Nevertheless, STING plays an important role in restricting the viral colonization of lymph nodes during secondary viremia.

## Methods

**Mice**. B6.129P2-Myd88tm1Aki x C57BL/6J-Ticam1Lps2 x B6.STOCK -Mavs (tm1Tsc) ($Myd88^{-/-}Trif^{-/-}Mavs^{-/-}$, MyTrMa KO)[60] and B6(Cg)-Tmem173tm1.2Camb/J ($Tmem173^{-/-}$, STING KO)[61] or B6(C)-Mb21d1tm1d (EUCOMM)Hmgu/J ($Cgas^{-/-}$, cGAS KO)[24] were intercrossed to generate

$Myd88^{-/-}Trif^{-/-}Mavs^{-/-}Tmem173^{-/-}$ (MyTrMaSt KO) or $Myd88^{-/-}Trif^{-/-}Mavs^{-/-}Cgas^{-/-}$ (MyTrMaGa KO) mice, respectively. In addition, $Tmem173^{-/-}$ mice were intercrossed with $Ifnb^{wt/\Delta\beta-luc}$ mice[41] in a manner that they carry one IFN-β$^{\Delta\beta-luc}$ allele, and C57BL/6-Siglec1<tm1(cre)Mtaka> ($CD169Cre^{+/-}$)[48] mice were intercrossed with B6.Ifnb(tm2(luc))Gbf-Bruce4 ($Ifnb^{flox-βluc}$)[41] to generate CD169-specific IFN-β reporter mice. B6.Cg-Tg(Alb-cre)21Mgn ($AlbCre^{+/-}$)[62] and B6.129P2-Lyz2tm1(cre)Ifo ($Lyz2Cre^{+/-}$)[63] were intercrossed with $Tmem173^{-/-}$ mice to generate $AlbCre^{+/-}Tmem173^{-/-}$ and $Lyz2Cre^{+/-}Tmem173^{-/-}$ mice. C57BL/6 (WT, Envigo), $Ifnar^{-/-}$[64], $Myd88^{-/-}Trif^{-/-}Mavs^{-/-}$, $Tmem173^{-/-}$, $Myd88^{-/-}Trif^{-/-}Mavs^{-/-}Tmem173^{-/-}$, $Cgas^{-/-}$, $Myd88^{-/-}Trif^{-/-}Mavs^{-/-}Cgas^{-/-}$, $Ifnb^{wt/\Delta\beta-luc}$, $Myd88^{-/-}Trif^{-/-}Mavs^{-/-}Ifnb^{wt/\Delta\beta-luc}$, $Tmem173^{-/-}Ifnb^{wt/\Delta\beta-luc}$, $Lyz2Cre^{+/-}Ifnb^{flox-βluc}$, $ItgaxCre^{+/-}Ifnb^{flox-βluc}$[65], $CD169Cre^{+/-}Ifnb^{flox-βluc}$, and $AlbCre^{+/-}Ifnb^{flox-βluc}$ mice were bred under specific pathogen-free conditions in the central mouse facility of the Helmholtz Centre for Infection Research, Brunswick, and at TWINCORE, Centre for Experimental and Clinical Infection Research, Hanover, Germany. Mouse experimental work was carried out using 8-week-old to 14-week-old mice in compliance with regulations of the German animal welfare law.

**Viruses**. To generate MCMVrep Δm157-flox-egfp, the flox-egfp cassette together with the FRT flanked kanamycin resistance marker was amplified by PCR using primers H5-Dm157-flox-egfp (5′-CGT GGT CAA GCC GGT CGT GTT GTA CCA GAA CTC GAC TTC GGT CGC GTT TTA ACG CTT ACA ATT TAC GGG GG-3′) and H3-Dm157-flox-egfp (5′-CCC CGA TAT TTG AGA AAG TGT ACC CCG ATA TTC AGT ACC TCT TGA CTA AGA AGC CAT AGA GCC CAC CGC-3′) and pCR3-FRT-kanr-FRT-flox-egfp[36] as template. The site-directed insertion of the PCR fragment into pSM3fr-MCK-2fl clone 3.3 (originally provided by Barbara Adler, Max von Pettenkofer Institute, LMU, Munich, GER[66]) into the m157 locus was performed by recombination with linear fragments[67] resulting in pSM3fr-MCK-2fl-Dm157-flox-egfp-FRT-kanr-FRT. The FRT flanked kanamycin fragment was removed by site-specific recombination[67] induced by pCP20 helper plasmid[68] resulting in pSM3fr-MCK-2fl Dm157-flox-egfp. To reconstitute the recombinant virus MCMVrep Δm157-flox-egfp, primary murine embryonic fibroblasts (pMEF) from BALB/c mice were transfected with purified pSM3fr-MCK-2fl-Dm157-flox-egfp DNA isolated from a sequence verified clone. The rescue-supernatant was passaged on pMEF six times to remove the BAC cassette from the virus genome[69] and virus stocks were propagated by infection of pMEF with the high passage inoculum of MCMVrep Δm157-flox-egfp.MCMV Δm157[70] (originally provided by Stefan Jordan, Icahn School of Medicine, Manhattan, NY), MCMV Δm157luc[43] (originally provided by Michael Mach, University Hospital Erlangen, Erlangen, GER), WT MCMV[66] (originally provided by Luka Cicin-Sain, Helmholtz Centre for Infection Research, Brunswick, GER) and MCK2-repaired MCMVrep Δm157-flox-egfp were propagated on pMEF from $C57BL/6$ or $BALB/c$ mice. Virus stocks were purified by sucrose density gradient ultracentrifugation. In all infection experiments mice were i.v. infected with $5 \times 10^5$ pfu MCMV. VACV strain Western Reserve (originally provided by Bernard Moss, NIH, Bethesda, MD) was propagated on BHK-21 cells. Infection was performed with $2 \times 10^6$ pfu VACV.

**Cytokine analyses**. Serum was tested for IFN-α and IFN-β using the VeriKine Mouse Interferon ELISA kits (PBL Assay Science) following the manufacturers' instructions.

**In vivo imaging**. Reporter mice were i.v. injected with 150 mg/kg of D-luciferin diluted in PBS (Intrace Medical SA), anesthetized with 2.5% Isofluran (Abbot), and monitored using an IVIS Spectrum CT (PerkinElmer). Photon flux was quantified using the Living Image 4.5.4 software (Caliper). For ex vivo luciferase activity measurements tissues were homogenized in BrightGlo lysis buffer (Promega) using a FastPrep24 homogenizer (MP Biomedicals). Lysates were mixed with a BrightGlo luciferase assay substrate (Promega), and luminescence was quantified using a Synergy 2 microplate reader (BioTek).

**Dissemination analysis and plaque assay**. Mice were infected with $5 \times 10^5$ pfu MCMV Δm157 or MCMVrep Δm157-flox-egfp. At the days of interest after infection mice were perfused with PBS, organs were removed and organ homogenates were plated in serial log10 dilutions on pMEF. Centrifugal enhancement was performed (2000 rpm, 15 min) and the cells were overlaid after 2 h of incubation at 37 °C and 5% $CO_2$ with 1% methylcellulose. Plaques were counted after 5 days of culture under a light microscope (Zeiss) and a fluorescence microscope (Nikon) or following staining with crystal violet.

**Statistical analysis**. Statistical analysis was carried out using Log-rank (Mantel Cox) test for survival analysis or a two-tailed Mann-Whitney test for all other comparisons. Values with $p < 0.05$ were considered statistically significant. Statistical analyses were performed using GraphPad Prism 5 software. Heatmaps were generated from mean values of each parameter using GraphPad Prism 7.

**Ethics statement**. All animals were handled in accordance with German animal welfare law and experiments were approved by the Niedersächsisches Landesamt

für Verbraucherschutz und Lebensmittelsicherheit (Oldenburg, Germany, identi-fication number 12/1025 and 17/2635).

**Reporting summary**. Further information on research design is available in the Nature Research Reporting Summary linked to this article.

## Data availability

The datasets generated during the current study are available from the corresponding author on reasonable request. The source data underlying Figs. 1a–d, 2a–b, 3b–e, 4b/d, 5b, 5d–e, and 6c–d, and Supplementary Figs. 1a–d, 2a–b, 3b and 4a–c are provided as a Source Data file.

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

## Acknowledgements

We thank Stefan Jordan (Icahn School of Medicine at Mount Sinai, New York) for providing MCMV Δm157 and Martin Messerle (Hanover Medical School, Hanover) for providing MCMV Δm157luc and WT MCMV. We thank Shizuo Akira (Osaka University, Osaka) for providing $Myd88^{-/-}$ mice, Bruce Beutler (UT Southwestern, Dallas) for providing $Trif^{-/-}$ mice, Charles M. Rice (The Rockefeller University, New York) for providing $Cgas^{-/-}$ mice, and Melanie Brinkmann for giving $Tmem173^{-/-}$ and $Tmem173^{+/-}Ifnb^{wt/\Delta\beta-luc}$ mice. We thank Lena Busker for experimental support. This study was supported by funding from the Helmholtz Association (Zukunftsthema "Immunology and Inflammation" (ZT-0027)) and funding from the Deutsche Forschungsgemeinschaft (SFB900, project B2, project number 158989968; Joint French-German Project cGAS-VAC, project number 406922110). The funders had no role in study design, data collection and analysis, decision to publish, or preparation of the manuscript.

## Author contributions

Conceptualization: P.K.T., J.S., and U.K.; methodology: P.K.T., J.S., A.R., S.L., Z.R., and U.K.; investigation: P.K.T., J.S., K.B., J.B., M.D., C.H., J.S.K., K.B., and L.G.; resources: S.L. and Z.R.; writing–original draft: P.K.T., J.S., and U.K., writing–review and editing: P.K.T., J.S., and U.K.; supervision: Z.R. and U.K.; funding acquisition: Z.R. and U.K.

## Additional information

**Competing interests:** The authors declare no competing interests.

