## [Peer Review File · Nature Communications]

Reviewers' comments:

Reviewer #1 (Remarks to the Author):

Tegtmeyer et al have carefully studied the two-phase induction of interferon during infection by the natural mouse pathogen, MCMV, which peak at 4-6 hpi and 36 hpi. Many DNA viruses activate cGAS-STING, although few of these are natural pathogens in the animals or cells being studied. There is little question that type I IFN(beta) contributes to restricting the first round of MCMV replication in vivo, but the constellation of pathogen sensors responsible for this activation is unknown. Here, they implicate combined STING plus MyD88/TRIF-dependent TLR plus MAVS-dependent RIG-I-like pathways in this process. They make the point that salivary gland-derived, MCK2-expressing MCMV was used, although they employed viruses that do not mediate a strong m157-specific Ly49H immune response seen with WT MCMV.

Importantly, neither cGAS- nor STING-deficient mice shows any increased susceptibility to MCMV even though deficiency in adaptors such as MyD88, TRIF, IRF3 and MAVS, like deficiency in type I interferon signaling, is well known to result in increased susceptibility to systemic infection.

Right off, they show clearly that the different phases exhibit different dependencies, with STING impacting the early activation (in apparent agreement with ref 34); whereas, MyD88/TRIF plus MAVS eliminated the second wave, implicating TLR and RIG-I-like signaling independent of cGAS-STING.

(In 201) They go on to nicely demonstrate that liver Kupffer cells (abbreviated K, but only in the middle of the text at line 207), known to be a target of MCMV, show a highly STING-dependent early response and that hepatocytes do not participate. In the text supporting these points, it would be helpful if the authors could mention any effort to see what happens in endothelial cells and how they distinguish these from Kupffer cells because both are established host of MCMV whereas hepatocytes have not been implicated as target cells for this infection.

(In 211) Likewise, authors need to complete the text with better distinction between CD11c+ DCs and Macs, if possible, even though LysM (activated Macs) and CD169 (Siglec-1 lectin) did not colocalize. The results are what they are, but the conclusions must be more objective.

Finally, the authors must show some data using WT (m157+) early in the experimental series, simply for completeness. It is not an issue regarding the excellent data shown, but the WT behavior is important to keep in mind, particularly because much of the Discussion does not continue to remind the reader of this important limitation of the study.

Dissemination following intraperitoneal inoculation with 5×10^5 PFU follows some but not all of the rules seen with routes of inoculation that may reflect more natural conditions, as pointed out in a number of manuscripts focused on dissemination over the past two decades. It may be that the liver is a dead end because of the route of inoculation, and the role of STING is a consequence of this unnatural route. When this virus enters its host by a more natural route where seeding of the spleen and liver is dependent upon dissemination rather than inoculum, the requirement for STING might be expected to be different.

All-in-all, the authors make a strong argument for the role of STING in the initial control of virus in the liver and spleen by a dose and route of inoculation that is common in experimental infections, organs that they point out are sites of initial replication following intraperitoneal inoculation. Virus from liver, in particular, does not seed (disseminate to) the rest of the host, and, importantly, STING is shown to not impact dissemination, so the picture is consistent.

Reviewer #2 (Remarks to the Author):

The authors report that STING or cGAS KO mice exhibit similar if not more resistance to MCMV

infection, but TLR/RLR/STING(cGAS) mice failed to mount appropriate innate defenses against fatal MCMV infection. However, STING does mediate early IFN- β induction in Kupffer cells which contribute to early virus propagation within the liver.

STING constrains MCMV replication only in myeloid cells to limit viral dissemination from these cells, but it failed to restrict viral dissemination from hepatocytes to other organs. The phenotypes were interesting and well supported by the data. However, the manuscript is descriptive and lacks mechanistic advances. Additionally, STING has been previously shown to play an important role in MCMV infection in vivo (PMID: 27334590) and thus, the impact of the study is not very high.

Major comments:

1. line 187-190, I do not agree with the description of Figure 3c, regarding "At 48 hpi very similar splenic BLI signals were detected in all analyzed mouse strains (Fig. 3A-D)". Clearly in Figure 3D, at 48 hpi, the expression among these three mice strains are quite different. The authors should show the actual values by including an extra table indicating the real numbers for Figure 3c and 3d.

2. In Figure 6, the authors should also infect LysMCre^{+/-}, LysMCre^{+/-}STING^{-/-}, AlbCre^{+/-} and AlbCre^{+/-}STING^{-/-} with luciferase expressing MCMV, and monitor the virus expanding status as in Figure 5A-C.

3. As the authors stated, STING plays important role in constraining MCMV replication in myeloid cells, which is responsible for viral dissemination. Can the authors reintroduce STING into the STING^{-/-} myeloid cells, and see if this will restrict MCMV replication?

4. The MyTRCaSt^{-/-} or MyTRCaGa^{-/-} mice exhibit significant susceptibility to MCMV infection. Because these mice lack protection from multiple innate immune pathways, it is likely that multiple viruses are affected. Could the authors include controls with another virus?

Minor comments:

1. The authors should consider labelling "liver" and "spleen" in figure 3c and 3d to make it easier for the reader.

Point-by-point reply to the reviewers' comments concerning the manuscript entitled "STING induces early IFN- β in the liver and constrains myeloid cell-mediated murine cytomegalovirus dissemination" by Tegtmeyer et al.

NCOMMS-18-26034

Reviewer #1:

1. (In 201) They go on to nicely demonstrate that liver Kupffer cells (abbreviated K, but only in the middle of the text at line 207), known to be a target of MCMV, show a highly STING-dependent early response and that hepatocytes do not participate. In the text supporting these points, it would be helpful if the authors could mention any effort to see what happens in endothelial cells and how they distinguish these from Kupffer cells because both are established host of MCMV whereas hepatocytes have not been implicated as target cells for this infection.

We appreciate the comment of reviewer #1 and rephrased the manuscript to further clarify the role of endothelial cells (EC) in MCMV infection and the discrimination of EC and Kupffer cells (KC). Indeed, EC are described as target cells for acute and latent CMV infection^{1,2}. In the liver they even show a higher initial infection rate when compared with hepatocytes and KC, which probably is due to their localization at the liver sinusoids³. The capability of EC to produce IFN-I following CMV infection was previously shown in experiments with isolated liver sinusoidal endothelial cells⁴ and with human umbilical vein endothelial cells, the latter of which showed a cGAS-STING-dependent induction of IFN-I responses⁵. To determine the IFN- β expression by Kupffer cells we used CD169-specific IFN- β reporter mice. CD169 is a lectin-like receptor, which is primarily expressed on tissue-resident macrophages. In immunofluorescent analysis of CD169 and the EC marker CD31 no colocalization was detected^{6,7}. Therefore, the possibility that also EC were targeted in the CD169-specific IFN- β reporter mice could be excluded. As these mice show a similar bioluminescent imaging signal as the ubiquitous IFN- β reporter mice that inform on the IFN- β induction of all cell types, we concluded that Kupffer cells were the major IFN- β producers during early MCMV infection. Based on the *in vitro* data mentioned above it is likely that also EC contributed to early IFN- β responses, however, under *in vivo* conditions their involvement can be only marginal. We extensively discussed the potential involvement of endothelial cells in the early IFN- β production in the revised manuscript (see line 362-378).

Regarding the role of hepatocytes, we (line 288-290) and others^{1,3} showed that hepatocytes are indeed direct target cells of a MCMV infection. Interestingly, although hepatocytes are readily infected and are able to produce IFN- β upon infection with other viruses, such as coxsackie virus B3⁸, MCMV infected hepatocytes do not significantly contribute to hepatic IFN- β responses as shown in Figure 4a/b of the original and revised manuscript. We rephrased the revised manuscript to highlight the role of hepatocytes as primary targets of MCMV infection and their inability to mount significant early IFN- β responses (line 351-362).

2. (In 211) Likewise, authors need to complete the text with better distinction between CD11c+ DCs and Macs, if possible, even though LysM (activated Macs) and CD169

(Siglec-1 lectin) did not colocalize. The results are what they are, but the conclusions must be more objective.

We thank reviewer #1 for this comment and agree that a clear distinction of CD11c⁺ dendritic cells (DC) and macrophages is important. The CD11cCre^{+/−} mice we used show high recombination efficiency in splenic DC, including conventional DC and plasmacytoid DC (pDC)^{9,10}. Cre expression under the CD11c promoter also induces recombination in red pulp macrophages (RPM) and marginal zone macrophages (MZM) of the spleen, whereas in these cell types the recombination is less efficient than in splenic DC subsets^{9,10}. Thus, the higher recombination efficiency in DC of CD11cCre mice and the earlier reports of the TLR-dependent sensing of MCMV by pDC at 36 hpi in the spleen led us to conclude that mainly CD11c⁺ DC contributed to the IFN-β expression at 36 hpi in the spleen (see Fig. 4c/d of the original and revised manuscript). The RPM and MZM are also targeted efficiently by the CD169Cre^{+/−} mouse line we used, whereas in LysMCre^{+/−} mice the recombination in these cell subsets was less efficient^{7,9}. This might explain the moderate bioluminescence differences in the spleen of MCMV infected LysMCre^{+/−} and CD169Cre^{+/−} mice (see Fig. 4c/d of the original and revised manuscript). We rephrased the manuscript accordingly in order to better distinguish CD11c⁺ DC and macrophages (line 213-215 and 388-390).

3. Finally, the authors must show some data using WT MCMV (m157+) early in the experimental series, simply for completeness. It is not an issue regarding the excellent data shown, but the WT behavior is important to keep in mind, particularly because much of the Discussion does not continue to remind the reader of this important limitation of the study.

We thank reviewer #1 for this suggestion and agree that it is important to analyze whether the absence of m157 influences the activation of the innate immune responses apart from natural killer cells. Therefore, we performed a new experiment in which we infected AlbCre^{+/−} and AlbCre^{+/−}STING^{−/−} mice with 5×10^5 pfu WT MCMV¹¹ and analyzed the IFN-β concentration in the serum at 4 and 36 hpi. Similar to the infection with MCMV Δm157, we observed STING-dependent serum IFN-β responses at 4 hpi, whereas the responses were independent of STING signaling at 36 hpi. Thus, we concluded that the absence of the m157 gene did not influence the way how MCMV is sensed by the innate immune system (line 173-176). We included the newly generated data into the new Supplementary Figure 2 of the revised manuscript.

4. Dissemination following intraperitoneal inoculation with 5×10^5 PFU follows some but not all of the rules seen with routes of inoculation that may reflect more natural conditions, as pointed out in a number of manuscripts focused on dissemination over the past two decades. It may be that the liver is a dead end because of the route of inoculation, and the role of STING is a consequence of this unnatural route. When this virus enters its host by a more natural route where seeding of the spleen and liver is dependent upon dissemination rather than inoculum, the requirement for STING might be expected to be different. All-in-all, the authors make a strong argument for the role of STING in the initial control of virus in the liver and spleen by a dose and route of inoculation that is common in experimental infections, organs that they point out are sites of initial replication following intraperitoneal inoculation. Virus from liver, in

particular, does not seed (disseminate to) the rest of the host, and, importantly, STING is shown to not impact dissemination, so the picture is consistent.

We appreciate the comment of reviewer #1 and agree that after local infection seeding of liver and spleen relies on virus dissemination rather than on the originally inoculated virus. The role of hepatocytes after local MCMV infection has already been addressed by Sacher et al.¹. In addition to the systemic infection route these authors also infected AlbCre^{+/-} mice intra-footpad and intranasally with the floxed reporter MCMV and analyzed the dissemination of hepatocyte-derived MCMV particles. Even after local infection and γ -irradiation MCMV was trapped in the liver. This implies that the dissemination blockade from the liver is independent of the route of infection. Furthermore, the physiological infection route for adult mice could not be clarified so far. Pups are reported to acquire MCMV infection primarily via inhalation, whereas adult mice show low level lung colonization after intranasal infection¹².

Reviewer #2:

1. line 187-190, I do not agree with the description of Figure 3c, regarding “At 48 hpi very similar splenic BLI signals were detected in all analyzed mouse strains (Fig. 3A-D)”. Clearly in Figure 3D, at 48 hpi, the expression among these three mice strains are quite different. The authors should show the actual values by including an extra table indicating the real numbers for Figure 3c and 3d.

We thank reviewer #2 for this comment and agree that the description of Figure 3 might be misleading. Therefore, we reformatted Figure 3 and added dot plots for the critical time points of the *ex vivo* IFN- β induction analysis (see Fig. 3e of the revised manuscript). The graphs comprise 0 and 4 hpi for liver and 0, 36, and 48 hpi for spleen. Additionally, we included statistics for the selected time points and rephrased the description of Figure 3 in the revised manuscript. It is now stated “In MyTrCa^{-/-}IFN- β ^{wt/ Δ Bluc} mice the splenic BLI signal increased between 36 and 48 hpi, nevertheless, compared with WT controls the signal intensity was still reduced at 48 hpi. In contrast, at that time STING^{-/-}IFN β ^{wt/ Δ Bluc} mice showed similar splenic BLI signals as WT controls (Fig. 3a-e)” (line 196-199).

2. In Figure 6, the authors should also infect LysMCre^{+/-}, LysMCre^{+/-}STING^{-/-}, AlbCre^{+/-} and AlbCre^{+/-}STING^{-/-} with luciferase expressing MCMV, and monitor the virus expanding status as in Figure 5A-C.

We thank reviewer #2 for this suggestion and infected LysMCre^{+/-}, LysMCre^{+/-}STING^{-/-}, AlbCre^{+/-}, and AlbCre^{+/-}STING^{-/-} mice with 5×10^5 pfu MCMV Δ m157luc and monitored virus expansion via *in vivo* imaging. Overall similar data were received in the *in vivo* imaging analysis and the plaque assays performed in Figure 6 (line 273, 282). We included the obtained data into the new Supplementary Figure 4 of the revised manuscript.

3. As the authors stated, STING plays important role in constraining MCMV replication in myeloid cells, which is responsible for viral dissemination. Can the authors reintroduce STING into the STING^{-/-} myeloid cells, and see if this will restrict MCMV replication?

We agree with reviewer #2 that reintroduction of STING into STING^{-/-} myeloid cells in order to test whether then MCMV replication is again restricted would be interesting. Unfortunately, this experiment is extremely difficult to carry out in an *in vivo* setting. Indeed, we believe that the data presented in the revised version of our manuscript clearly support the hypothesis that STING is needed to restrict MCMV replication in myeloid cells *in vivo* (see Fig. 6c of the original and revised manuscript, titers of MCMVrec in the liver on day 3 and in the lymph nodes on day 8).

4. The MyTrCaSt^{-/-} or MyTrCaGa^{-/-} mice exhibit significant susceptibility to MCMV infection. Because these mice lack protection from multiple innate immune pathways, it is likely that multiple viruses are affected. Could the authors include controls with another virus?

We agree with reviewer #2 that our MyTrCaSt^{-/-} and MyTrCaGa^{-/-} mice show deficiencies in multiple innate sensing pathways that presumably render these mice susceptible to a variety of viral and bacterial pathogens. While upon MCMV infection the deficiency of all three signaling platforms, including TLR, RLR, and cGAS/STING, as realized in MyTrCaSt^{-/-} and MyTrCaGa^{-/-} mice, was needed to render the mice susceptible to lethal MCMV infection, deletion of STING alone was sufficient to enhance the susceptibility to lethal infection with the alpha-herpesvirus HSV-1¹³⁻¹⁵. This phenotype was reminiscent of that detected after VACV infection (see Supplementary Fig. 1). Thus, it is noteworthy that during MCMV infection redundant TLR, RLR and STING signaling confers the induction of protective immunity, whereas during HSV-1 and VACV infection STING signaling alone plays a critical role.

Minor comments:

1. The authors should consider labelling “liver” and “spleen” in figure 3c and 3d to make it easier for the reader.

We thank reviewer #2 for this advice and included the labeling “Liver” and “Spleen” in the revised Figure 3c/d.

- 1 Sacher, T. *et al.* The major virus-producing cell type during murine cytomegalovirus infection, the hepatocyte, is not the source of virus dissemination in the host. *Cell Host Microbe* **3**, 263-272, doi:10.1016/j.chom.2008.02.014 (2008).
- 2 Seckert, C. K. *et al.* Liver sinusoidal endothelial cells are a site of murine cytomegalovirus latency and reactivation. *J Virol* **83**, 8869-8884, doi:10.1128/JVI.00870-09 (2009).
- 3 Lemmermann, N. A. *et al.* Non-redundant and redundant roles of cytomegalovirus gH/gL complexes in host organ entry and intra-tissue spread. *PLoS Pathog* **11**, e1004640, doi:10.1371/journal.ppat.1004640 (2015).
- 4 Kern, M. *et al.* Virally infected mouse liver endothelial cells trigger CD8+ T-cell immunity. *Gastroenterology* **138**, 336-346, doi:10.1053/j.gastro.2009.08.057 (2010).
- 5 Lio, C. W. *et al.* cGAS-STING Signaling Regulates Initial Innate Control of Cytomegalovirus Infection. *J Virol* **90**, 7789-7797, doi:10.1128/JVI.01040-16 (2016).
- 6 Karasawa, K. *et al.* Vascular-resident CD169-positive monocytes and macrophages control neutrophil accumulation in the kidney with ischemia-reperfusion injury. *J Am Soc Nephrol* **26**, 896-906, doi:10.1681/ASN.2014020195 (2015).

- 7 Gupta, P. *et al.* Tissue-Resident CD169(+) Macrophages Form a Crucial Front Line against Plasmodium Infection. *Cell Rep* **16**, 1749-1761, doi:10.1016/j.celrep.2016.07.010 (2016).
- 8 Koestner, W. *et al.* Interferon-beta expression and type I interferon receptor signaling of hepatocytes prevent hepatic necrosis and virus dissemination in Cocksackievirus B3-infected mice. *PLoS Pathog* **14**, e1007235, doi:10.1371/journal.ppat.1007235 (2018).
- 9 Abram, C. L., Roberge, G. L., Hu, Y. & Lowell, C. A. Comparative analysis of the efficiency and specificity of myeloid-Cre deleting strains using ROSA-EYFP reporter mice. *J Immunol Methods* **408**, 89-100, doi:10.1016/j.jim.2014.05.009 (2014).
- 10 Caton, M. L., Smith-Raska, M. R. & Reizis, B. Notch-RBP-J signaling controls the homeostasis of CD8- dendritic cells in the spleen. *J Exp Med* **204**, 1653-1664, doi:10.1084/jem.20062648 (2007).
- 11 Jordan, S. *et al.* Virus progeny of murine cytomegalovirus bacterial artificial chromosome pSM3fr show reduced growth in salivary Glands due to a fixed mutation of MCK-2. *J Virol* **85**, 10346-10353, doi:10.1128/JVI.00545-11 (2011).
- 12 Farrell, H. E. *et al.* Murine Cytomegalovirus Exploits Olfaction To Enter New Hosts. *MBio* **7**, e00251-00216, doi:10.1128/mBio.00251-16 (2016).
- 13 Parker, Z. M., Murphy, A. A. & Leib, D. A. Role of the DNA Sensor STING in Protection from Lethal Infection following Corneal and Intracerebral Challenge with Herpes Simplex Virus 1. *J Virol* **89**, 11080-11091, doi:10.1128/JVI.00954-15 (2015).
- 14 Li, X. D. *et al.* Pivotal roles of cGAS-cGAMP signaling in antiviral defense and immune adjuvant effects. *Science* **341**, 1390-1394, doi:10.1126/science.1244040 (2013).
- 15 Ishikawa, H., Ma, Z. & Barber, G. N. STING regulates intracellular DNA-mediated, type I interferon-dependent innate immunity. *Nature* **461**, 788-792, doi:10.1038/nature08476 (2009).

REVIEWERS' COMMENTS:

Reviewer #1 (Remarks to the Author):

The authors have done an admirable job if revising the ms in response to the concerns of reveiwers.

Reviewer #2 (Remarks to the Author):

The authors have answered the previous concerns.